# Rootstocks Impact Yield, Fruit Composition, Nutrient Deficiencies, and Winter Survival of Hybrid Cultivars in Eastern Canada

**Caroline Provost \*, Alexander Campbell and François Dumont**

Centre de Recherche Agroalimentaire de Mirabel, 9850 Rue Belle-Rivière, Québec, QC J7N 2X8, Canada; acampbell@cram-mirabel.com (A.C.); fdumont@cram-mirabel.com (F.D.)
\* Correspondence: cprovost@cram-mirabel.com; Tel.: +1-450-434-8050 (ext. 6064)

**Abstract:** Grafting cold-hardy hybrid grapevines may influence their attributes under different pedoclimatic conditions and may also contribute to cold-hardiness, influence plant physiology, and affect yield and fruit composition. In a six-year study, we evaluated bud survival, plant development, nutrient deficiencies, yield, and fruit composition for three cold-hardy grape varieties: Frontenac, Frontenac blanc, and Marquette. The grape varieties were grafted on four rootstocks: 3309C, SO4, Riparia Gloire, and 101-14. The final combinations were own-rooted. The six-year research period indicated that cold-hardy hybrids were affected differently by each rootstock. Magnesium deficiency was lower for grafted Frontenac and Frontenac blanc compared with own-rooted vines, but bud survival and grapevine development were not affected by rootstock. Moreover, results related to yield components showed that there are significant differences between rootstocks and own-rooted vines. Frontenac was the least affected grape variety compared to Frontenac blanc and Marquette, where only cluster weight and berry weight were impacted. Overall, for the two Frontenac varietals, we also observed a greater maturity for fruits of vines grafted on 101-14 and 3309C compared with own-rooted vines. Grafting affected fruit composition for Marquette differently, where the lowest grape maturity was observed for fruits on vines grafted on SO4. This study demonstrates that rootstocks affect cold-hardy hybrids, highlighting their potential under eastern North American conditions.

**Keywords:** cold-hardy hybrid; rootstock effect; cold climate; Frontenac; Marquette





## 1. Introduction

Growing grapes in cold climates presents several challenges to overcome. Grape production is a relatively recent industry in eastern Canada, and growers must adapt their techniques to achieve high grape quality at the end of the season. Cold injury to grapevines, short growing seasons, and soil conditions that are often too fertile and poorly drained are just a few examples of factors that affect grapevine production and limit the choice of grape variety when establishing a vineyard [1,2]. Pedoclimatic conditions found in Quebec (eastern Canada) limit the choice of grape varieties that can be used; therefore, winegrowers are often restricted to grape varieties tolerant to extreme winter temperatures and spring frosts, but also those capable of reaching optimum berry maturity at the end of the season [2]. While genetics determines the ultimate degree of cold-hardiness expression, the environment, as well as cultural practices and pest management, affect that expression. Cold injury and disease resistance studies have helped lead breeding programs in eastern North America and have led to the development of several hybrid cultivars with high cold hardiness [3].

Rootstocks allow growers to plant varieties that become better adapted and more efficient in specific soil and climatic conditions and improving the scion/rootstock combination optimizes their adaptation. Though some cold-hardy hybrids have been grafted and studied, this is not a common practice for hybrid cultivars as rootstocks are typically used

for *Vitis vinifera*. In recent years, some studies have been carried out in British Columbia (Canada), Ontario (Canada), New York (USA), and Missouri (USA) to assess the benefits of using rootstocks for hybrid grape varieties grown in cold climates [4–10].

Some studies have noted that rootstocks can influence scion cold-hardiness through faster cold acclimation periods [6,11]. Other studies have observed no difference in cold-hardiness according to rootstock across multiple scion/rootstock combinations [12,13]. Rootstocks can also influence vine vigor since it is the root system that provides the plant with water and mineral uptake essential to its growth and harbors the majority of nutritional reserves that are stored during the winter season [14]. Some rootstocks enhance the physiological development of the vine and can ensure optimal ripening of the grafted grape variety [7,14,15]; thus, rootstocks also have an impact on yield and berry quality. Studies have shown that there is a significant interaction between grape varieties and rootstocks related to yield, the accumulation of sugars in berries, the chemical composition of the berries, and the aromas [5,7,15,16]. Others have observed more variable results, where grafted plants are often similar to those that are own-rooted [15].

Grapevine grafting may influence grape production in specific soil and climate conditions, specifically in an emergent grapevine production region such as Quebec. This project evaluated the effects of grafting on cold-hardiness, grapevine development, grape maturation, yield, and berry chemistry of cold-hardy hybrids. These results can be used by growers, stakeholders, nurseries, and researchers for development of the wine industry in eastern Canada.

## 2. Materials and Methods

All trials were conducted at the experimental vineyard of the Centre de recherche agroalimentaire de Mirabel (CRAM) located in the municipality of Oka, Quebec, Canada (45°30′ N, 74°4.2′ W). The region's climate is characterized by cold winters with temperatures reaching as low as $-30\,°C$, short growing seasons determined in part by days without frost (generally 155 to 200 days calculated from the last spring frost to the first fall frost), and also hot summers that accumulate 1295–1450 growing degree days (base 10, mean of the last 10 years, 1365 GDD) (Table 1). The climate is also humid, with mean annual precipitation of 790 mm of rain during the growing season (snow excluded). The vineyard soil type is a gravelly loam, and plots are located on an 8% slope.

**Table 1.** Characteristics of grape varieties.

| Scion | Origin | Hardiness | GDD at Harvest (Base 10) | Vigor | Mean Yield | Deficiency |
|---|---|---|---|---|---|---|
| Frontenac | *Vitis riparia* 89 × Landot 4511 | $-30$ to $-34\,°C$ | 1250 | High | 8 to 12 T ha$^{-1}$ | Mg |
| Frontenac blanc | Mutation from the Frontenac | $-30$ to $-34°C$ | 1150 | High | 8 to 12 T ha$^{-1}$ | Mg |
| Marquette | Mn 1094 × Ravat 26,212 | $-30$ to $-34°C$ | 1100 | High | 6 to 10 T ha$^{-1}$ | |

Sources: [1,17,18].

Mature dormant canes of three cold-hardy hybrids were collected in October 2011 from the CRAM vineyard and provided to a commercial nursery for bench grafting. Vines were grown for a year in the nursery (2012) and shipped for planting in spring 2013 as dormant vines. Grafted vines were planted in June 2013. The experimental design is composed of four replicates (plots) for each of the scion/rootstock combinations (3 grape varieties, 5 root systems). Three cold-hardy hybrid varieties were used: Frontenac, Frontenac blanc, and Marquette (Table 1), and five root systems were evaluated: Couderc 3309 (3309C), Sélection Oppenheim 4 (SO4), Riparia Gloire (RP), Millardet et de Grasset 101-14 (101-14), and own-rooted (Table 2). Hybrid cultivars were chosen on the basis of yield and quality components as well as land use by local growers. Rootstocks were selected on the basis of qualities known to be imparted on scions as well as availability and grower selection.

Each plot included 10 grapevines, for a total of 40 vines per combination. The four blocks containing the combinations were implanted according to a complete random distribution. Rows were oriented North–South, and the vines were planted to 1.20 × 2.44 m spacing within and between the rows. Grapevines were trained to a bilateral cordon and vertical shoot positioning system. An initial pruning was performed during the month of April, and the final pruning was completed in May, after the risk of spring frost had passed, to leave 16 nodes per vine. In 2015, the first small harvest was collected. Data were collected from April 2014 to October 2019 (six growing seasons).

**Table 2.** Characteristics of rootstocks.

| Rootstock | Origin | Vigor | Soil Adaptation | Nutrient Absorption |
|---|---|---|---|---|
| Couderc 3309 (3309C) | *V. riparia* × *Vitis rupestris* | Medium | Adapted to various types of soils (deep, sandy-clay, silty-clay, little limestone) and well-drained. High tolerance to soil acidity. | Absorbs magnesium well, and low absorption of potassium |
| Sélection Oppenheim 4 (SO4) | *Vitis berlandieri* × *V. riparia* | Medium to high | Adapts to various types of soils (sandy, clay-limestone, medium or low fertility), best results in well-drained, moist, and rich soils. | Badly assimilates magnesium |
| Riparia Gloire (RP) | *V. riparia* | Low | All non-calcareous, rich, and fresh soils. Adapted to acidic soils. Not suitable for too clayey and compact soils. | Easily absorbs potassium and low absorption of magnesium |
| Millardet et de Grasset 101-14 (101-14) | *V. riparia* × *V. rupestris* | Medium | Deep and clayey soils. Sensitive to soil acidity. | Absorbs magnesium well |

Sources: [9,17,19].

The between row vineyard floor consisted of a permanent cover crop, and herbicides were used under the row (three treatments per year) to control weeds Canopy management practices (hedging, shoot positioning, shoot thinning, leaf removal) were performed all season according to integrated pest management practices. To prevent severe infestations and heavy losses in the plots due to disease and insect damage, we applied chemical sprays as recommended by an agronomist and followed integrated pest management practices.

*2.1. Viticultural Parameters*

Data were collected throughout the growing season and measured on the six central plants of each plot. Several parameters were monitored: (a) bud survival, where bud break was evaluated on 6 buds from two canes per vine to determine winter mortality; (b) grapevine phenology using the BBCH scale [20] with observations twice a week from early April to early June, and once a week from June to September; (c) nutrient deficiencies (e.g., nitrogen, magnesium, manganese) were evaluated at three time points during the growing season using the Horsfall–Barratt grade scale [21]; (d) grapevine vigor was evaluated once a year (in July) using three parameters: shoot length on two canes, trunk diameter, and leaf area index (LAI) (Li-Cor LAI-2200C Plant Canopy Analyzer). At the end of the growing season, berry maturity was followed (pH, total acidity (TA)and total soluble solids (TSS)) using weekly chemical analyses on 100 berries from the end of August to the end of September. The optimum harvest times were determined using maturity monitoring. Measurements taken at harvest include yield as kilograms per vine, clusters per vine and berry weight, calculated from random sampling of 100 berries within the sampled vines. Cluster weight was then determined from these data. Chemical composition of fruit at harvest was analyzed using a random sampling of 200–300 berries. All the chemical analyses were performed on fresh juice by using a wine titrator for titratable acidity (Hanna, model

HI 84102), a refractometer for soluble solids (Hanna, model HI96811), and a pH-meter (Hanna, model HI9124).

### 2.2. Statistical Analysis

The results for the different parameters were analyzed separately for each grapevine variety. A generalized linear mixed model (GLMER) for binomial distribution was implemented to test the effect of rootstocks, year, and the interaction of rootstock x year on bud survival related to winter injuries. The plot nested in the year was included in the model. The statistical significance of the fixed effect was estimated using a likelihood ratio test (LRT) for chi-squared distribution. The levels of nutrient deficiencies were tested using a GLMER model for Poisson distributed data to test the effect of rootstocks, year, and the interaction of rootstock $\times$ year. The fixed effect rootstock and the random effect plot nested in the year were included in the model. A Tukey's test using the glht function in R (package multcomp) was used for comparison among treatments. Data related to harvest components and fruit composition were analyzed using two-way ANOVA with rootstock, year, and interaction as fixed parameters. The analyses were performed in R (R Core Team 2017) using the functions of the lme4 library.

### 3. Results

The results presented are the first, and relevant conclusions will come in the future with additional years of observations. Data related to cold-hardiness, developmental stages, and nutrient deficiencies are presented as the average value for six years and, for data related to yield and fruit composition, as the average for five years, from the first harvest in 2015 until 2019. The annual growing season effect was observed in years with shorter or longer than normal GDD accumulation (Table 3). The shortest growing season was observed in 2018 with 1253 GDD and 177 days without frost; the seasons with the highest GDD being noted in 2015 and 2016.

**Table 3.** Climatic conditions in Oka vineyard during 2014 to 2019.

| Year | GDD (Base10) | Precipitations (mm) | Days without Frost |
|------|--------------|---------------------|--------------------|
|      | **1 March to 30 October** | **1 March to 30 October** |  |
| 2019 | 1305 | 630 | 186 |
| 2018 | 1253 | 550 | 177 |
| 2017 | 1373 | 879 | 190 |
| 2016 | 1424 | 901 | 179 |
| 2015 | 1429 | 923 | 190 |
| 2014 | 1395 | 450 | 202 |

### 3.1. Bud Survival

Bud survival was not significantly affected by rootstock (Table 4). Statistical analysis showed that bud survival after winter for the three cold-hardy hybrids were similar among rootstock and own-rooted vines. Seasonal effects on bud survival across all three grape varieties were also observed; as the winters in 2014 and 2018 caused more bud damage than winters in 2015, 2016, 2017, and 2019.

### 3.2. Grapevine Development

Grafting had no effect on vine physiology during the spring nor for the remainder of the season, save for one exception during early season growth. In early May of 2017, we observed a faster development for Marquette vines grafted on Riparia Gloire and SO4 than for other rootstocks or own-rooted vines (data not shown). Two weeks later, no difference in the development of the vine was observed ($p = 0.0021$), and that tendency was not maintained in subsequent years. The BBCH stage 5 'wool stage' was observed for the two Frontenac varieties at the beginning of May, stage 9 'bud burst' around mid-May, flowering stage 65 near mid-June, and ripening at the beginning of August.

**Table 4.** Effect of rootstock on bud survival of the three cold hardy hybrids.

| | | Bud Survival (%) | | |
|---|---|---|---|---|
| | **Rootstock** | **Frontenac** | **Frontenac Blanc** | **Marquette** |
| Rootstock | 101-14 | 82.47 | 78.16 | 82.12 |
| | 3309C | 88.93 | 87.69 | 82.18 |
| | Own-rooted | 90.91 | 88.96 | 83.55 |
| | Riparia gloire | 86.93 | 87.69 | 86.88 |
| | SO4 | 87.25 | 86.32 | 83.59 |
| Year | 2014 | 66.25 | 63.25 | 67.11 |
| | 2015 | 90.63 | 89.27 | 88.05 |
| | 2016 | 90.63 | 89.71 | 89.82 |
| | 2017 | 97.36 | 97.81 | 96.96 |
| | 2018 | 84.06 | 80.80 | 71.71 |
| | 2019 | 91.67 | 91.26 | 84.87 |
| *p*-value | Rootstock | 0.83 | 0.42 | 0.91 |
| | Year | 0.19 | 0.15 | 0.64 |
| | Roostock × Year | 0.77 | 0.91 | 0.92 |

The grapevine vigor was affected by the rootstock and year (Table 5). For all the grapevine varieties, the trunk diameter increased yearly after plantation. For Frontenac and Frontenac blanc, shoot length and LAI were lower during summer in 2016 and 2017 than during other growing seasons. For Marquette, seasons 2015, 2016 and 2017 resulted in a lower shoot length and LAI. The rootstocks affected the Frontenac grapevine vigor where grating on Riparia Gloire resulted in a lower shoot length, trunk diameter, and LAI. LAI was also weak for own-root vines. Vigor of Frontenac blanc was low on rootstock 101-14 and moderately low (trunk diameter and LAI) on Riparia Gloire. Vigor parameters showed different results for Marquette. Shoot length was lower on 3309C and 101-14, the trunk diameter was weak on Riparia Gloire and 101-14, and the lowest value of LAI was observed on Riparia Gloire.

**Table 5.** Effect of rootstock on grapevine vigor of the three cold hardy hybrids.

| | | Frontenac | | | Frontenac Blanc | | | Marquette | | |
|---|---|---|---|---|---|---|---|---|---|---|
| | **Rootstock** | **Shoot Length (cm)** | **Trunk Diameter (cm)** | **Leaf Area Index** | **Shoot Length (cm)** | **Trunk Diameter (cm)** | **Leaf Area Index** | **Shoot Length (cm)** | **Trunk Diameter (cm)** | **Leaf Area Index** |
| Rootstock | 101-14 | 189.94 | 17.82 [a] | 4.65 [ab] | 161.28 [a] | 16.50 [ab] | 4.17 [bc] | 170.81 [a] | 15.48 [bc] | 4.75 [abc] |
| | 3309C | 187.57 | 18.01 [a] | 4.90 [a] | 174.07 [ab] | 17.91 [a] | 4.49 [abc] | 169.84 [a] | 17.35 [a] | 5.01 [bc] |
| | Own-root | 186.60 | 17.56 [a] | 4.21 [c] | 190.78 [c] | 17.67 [a] | 4.80 [a] | 185.66 [b] | 17.24 [ab] | 4.62 [ab] |
| | Riparia Gloire | 174.71 | 14.92 [b] | 4.35 [bc] | 178.73 [bc] | 15.98 [b] | 4.08 [c] | 172.44 [a] | 14.88 [c] | 4.40 [a] |
| | SO4 | 183.73 | 16.93 [a] | 4.58 [abc] | 175.56 [b] | 17.05 [ab] | 4.61 [ab] | 188.27 [b] | 18.17 [a] | 5.23 [c] |
| Year | 2014 | 214.67 [a] | 9.26 [a] | | 188.60 [ab] | 8.95 [a] | | 220.31 [a] | 8.97 [a] | |
| | 2015 | 180.68 [bc] | 13.81 [b] | | 175.88 [bc] | 13.88 [b] | | 166.86 [c] | 13.87 [b] | |
| | 2016 | 169.04 [bc] | 16.20 [c] | 3.93 [a] | 166.03 [cd] | 16.61 [c] | 3.67 [a] | 166.03 [c] | 16.54 [c] | 4.00 [a] |
| | 2017 | 160.87 [c] | 19.15 [d] | 3.91 [a] | 159.36 [d] | 19.34 [d] | 4.02 [a] | 150.14 [d] | 17.81 [c] | 4.58 [b] |
| | 2018 | 198.23 [ab] | 21.69 [e] | 5.15 [b] | 192.59 [a] | 22.16 [e] | 5.04 [b] | 189.95 [b] | 21.51 [d] | 5.32 [c] |
| | 2019 | 184.99 [bc] | 21.77 [e] | 5.15 [b] | 174.05 [bcd] | 21.20 [e] | 5.04 [b] | 171.70 [c] | 21.62 [d] | 5.32 [c] |
| *p*-value | Rootstock | 0.4874 | <0.0001 | <0.0001 | <0.0001 | 0.0025 | 0.0001 | <0.0001 | <0.0001 | <0.0001 |
| | Year | <0.0001 | <0.0001 | <0.0001 | <0.0001 | <0.0001 | <0.0001 | <0.0001 | <0.0001 | <0.0001 |
| | Roostock × Year | 0.3332 | 0.3731 | 0.0581 | 0.0744 | 0.9167 | 0.4584 | 0.0013 | 0.1013 | 0.6048 |

Values followed with a different letters in each column (section 'Rootstock' and 'Year', separately) were significantly different according to ANOVA.

### 3.3. Nutrient Deficiencies

When present, nutrient deficiencies were recorded for magnesium, nitrogen, potassium, and manganese. No deficiency in potassium was observed, and only a few plants showed manganese deficiencies. Nitrogen deficiency was very uncommon as well—we observed between two and eight vines yearly affected by nitrogen deficiency, mainly those grafted to rootstock 101-14. The main nutrient deficiency noted was for magnesium on the two Frontenac varieties (Table 6). Vines of Frontenac showed higher magnesium deficiency for own-rooted vines and vines grafted on SO4 compared to vines grafted on Riparia Gloire, 101-14, and 3309C. The least magnesium-deficient vines were grafted on 3309C. For Frontenac blanc, low magnesium deficiency was observed for vines grafted on 3309C, intermediate deficiency was obtained on vines grafted on 101-14 and Riparia Gloire, and the highest level of magnesium deficiency was noted for own-rooted vines and vines grafted on SO4. The Marquette variety showed lower levels of magnesium deficiency than Frontenac, although, again, significantly higher deficiency was observed for vines grafted on SO4 than for vines grafted on 101-14 and 3309C. The growing season also affected magnesium deficiency.

**Table 6.** Effect of rootstock on nutrient deficiencies of the three cold hardy hybrids.

| | | Level of Magnesium Deficiency (Horsfall–Barratt Grade Scale) | | |
| --- | --- | --- | --- | --- |
| | Rootstock | Frontenac | Frontenac Blanc | Marquette |
| Rootstock | 101-14 | 2.22 [b] | 2.02 [b] | 0.79 [a] |
| | 3309C | 1.35 [a] | 1.49 [c] | 0.76 [a] |
| | Own-rooted | 3.59 [c] | 2.88 [ab] | 0.86 [ab] |
| | Riparia gloire | 2.66 [b] | 2.35 [b] | 1.07 [ab] |
| | SO4 | 3.51 [c] | 3.15 [a] | 1.78 [b] |
| Year | 2014 | 0.84 | 0.78 | 0.80 |
| | 2015 | 1.00 | 0.95 | 0.95 |
| | 2016 | 5.49 | 5.18 | 1.78 |
| | 2017 | 0.81 | 0.66 | 0.03 |
| | 2018 | 3.55 | 3.03 | 0.65 |
| | 2019 | 4.31 | 3.74 | 2.21 |
| *p*-value | Rootstock | <0.0001 | 0.0001 | 0.02 |
| | Year | 0.08 | 0.10 | 0.99 |
| | Roostock × Year | 0.01 | 0.006 | 0.25 |

Values followed with a different letters in each column (section 'Rootstock' and 'Year', separately) were significantly different according to GLMER model.

### 3.4. Yield Components and Fruit Composition

The five-year observation period showed different significant effects for rootstocks and growing season on yield and quality for the three grape varieties.

#### 3.4.1. Frontenac

Rootstocks affected some yield components and fruit composition parameters for Frontenac, and we observed a significant effect of the growing season (Table 7). Statistical analysis showed that the rootstocks affect cluster weight, berry weight, soluble solids, and pH. Vines grafted on Riparia Gloire produced the heaviest clusters, followed by own-rooted vines and vines grafted on 3309C, while the lightest clusters were produced on vines grafted to SO4. Across all combinations, cluster weight varied from 93.60 g to 105.01 g. Significant differences for berry weight were noted between rootstocks: the lowest berry weights were for vines grafted on 101-14 and 3309C, and the highest for own-rooted vines, ranging from 1.10 to 1.26 g. The number of clusters and yield were not significantly affected by rootstocks. The average number of clusters varied between 30 to 33 clusters, and each grapevine produced slightly more than 3 kg. Rootstocks affected fruit composition

for soluble solids and pH, although titratable acidity was not affected. Higher levels of soluble solids were seen for vines grafted on 101-14, 3309C, and Riparia Gloire compared to own-rooted vines. Results for pH followed the same trends as soluble solids.

**Table 7.** Effect of rootstock on yield quantity and fruit composition of grapevine 'Frontenac'.

| | Rootstock | Average Number of Cluster (pcs) | Cluster Weight (g) | Berry Weight (g) | Yield (kg·vine$^{-1}$) | Yield (t·ha$^{-1}$) | Soluble Solids (°Brix) | pH | Titratable Acidity (g·L$^{-1}$ tar. ac.) |
|---|---|---|---|---|---|---|---|---|---|
| | | | | | **Frontenac** | | | | |
| | 101-14 | 32.31 | 96.22 [bc] | 1.10 [a] | 3.16 | 10.66 | 24.77 [a] | 3.22 [a] | 12.89 |
| | 3309C | 32.78 | 101.72 [abc] | 1.14 [ab] | 3.33 | 11.22 | 24.23 [a] | 3.20 [a] | 13.20 |
| Rootstock | Own-rooted | 30.88 | 102.22 [ab] | 1.26 [c] | 3.19 | 10.74 | 21.98 [c] | 3.13 [b] | 13.91 |
| | Riparia Gloire | 30.75 | 105.01 [a] | 1.17 [b] | 3.44 | 11.58 | 23.94 [ab] | 3.18 [ab] | 13.26 |
| | SO4 | 31.58 | 93.6 [c] | 1.19 [b] | 3.04 | 10.25 | 22.88 [bc] | 3.18 [a] | 13.67 |
| | 2015 | 25.00 [a] | 85.62 [a] | 1.00 [a] | 2.16 [a] | 7.29 [a] | 24.23 [a] | 3.58 [a] | 11.18 [a] |
| | 2016 | 35.97 [b] | 106.41 [b] | 1.27 [c] | 3.77 [b] | 12.70 [b] | 24.73 [a] | 3.20 [b] | 13.21 [bc] |
| Year | 2017 | 29.75 [c] | 80.92 [a] | 1.13 [b] | 2.42 [ac] | 8.14 [ac] | 21.53 [b] | 3.12 [c] | 12.14 [ab] |
| | 2018 | 24.64 [a] | 106.36 [b] | 1.23 [c] | 2.66 [c] | 8.97 [c] | 23.73 [a] | 3.20 [b] | 13.91 [c] |
| | 2019 | 42.98 [d] | 119.70 [c] | 1.22 [c] | 5.15 [d] | 17.36 [d] | 23.76 [a] | 2.90 [d] | 15.95 [a] |
| | Rootstock | 0.4571 | 0.0010 | <0.0001 | 0.1150 | 0.1150 | <0.0001 | <0.0001 | 0.2419 |
| *p*-value | Year | <0.0001 | <0.0001 | <0.0001 | <0.0001 | <0.0001 | <0.0001 | <0.0001 | <0.0001 |
| | Roostock × Year | 0.4765 | 0.0602 | 0.3386 | 0.0268 | 0.0268 | 0.5083 | 0.2806 | 0.8173 |

Values followed with a different letters in each column (section 'Rootstock' and 'Year', separately) were significantly different according to ANOVA.

Growing season also affected yield components and fruit composition (Table 7). The highest yields were obtained during 2019 and 2016; the first harvest in 2015 was the lowest; yields during 2017 and 2018 were intermediate. We do not have a constant trend between years for the number of clusters, cluster weight, and berry weight. The number of clusters ranged from 24 to 43 clusters, and the lowest number of clusters was observed during 2015 and 2018. The high number of clusters noted for 2019 may be related to the high yield. The cluster weight varied from 80.92 g in 2017 to 119.70 g in 2019, and the berry weight was the lowest during 2015. The heaviest berries were noted during the 2016 growing season. Lastly, berry chemistry was also affected by growing season. Statistical analyses showed that soluble solids were similar for the growing seasons of 2015, 2016, 2018, and 2019, varying from 23.73 to 24.73 °Brix; only 2017 showed a lower sugar content with 21.53 °Brix. pH was lowest in 2019, and highest during 2015. Titratable acidity was very high during 2019, and the lowest value was seen for the first harvest in 2015.

### 3.4.2. Frontenac Blanc

The rootstocks had a greater impact on Frontenac blanc than on the Frontenac variety. We observed a significant effect of the rootstocks and growing season on all the studied parameters, except for the average number of clusters (Table 8). The cluster weight ranged from 87.64 to 110.70 g and differed significantly between rootstocks. Vines grafted on SO4 had significantly lighter clusters than those grafted on Riparia Gloire, 3309C and own-rooted vines. The heaviest clusters were observed on own-rooted vines. Statistical analyses showed significantly higher berry weight for own-rooted vines and vines grafted on Riparia Gloire compared to rootstocks 3309C, 101-14, and SO4. Yield ranged from 3.10 to 3.87 kg per vine and was highest for own-rooted vines, while vines grafted on 101-14 and SO4 recorded the lowest yields. Fruit composition was related to grape maturity, where the lowest grape maturity was observed for own-rooted vines with low soluble solid concentration and high titratable acidity. All vines grafted on the four rootstocks showed similar fruit composition at harvest, with soluble solid values ranging from 23.27 to 24.18 °Brix and titratable acidity between 13.06 and 13.63 g/L.

**Table 8.** Effect of rootstock on yield quantity and fruit composition of grapevine 'Frontenac blanc'.

| | | | | | | | | | |
|---|---|---|---|---|---|---|---|---|---|
| | **Frontenac Blanc** | | | | | | | | |
| | **Rootstock** | **Average Number of Clusters (pcs)** | **Cluster Weight (g)** | **Berry Weight (g)** | **Yield (kg·vine⁻¹)** | **Yield (t·ha⁻¹)** | **Soluble Solids (°Brix)** | **pH** | **Titratable Acidity (g·L⁻¹ tar. ac.)** |
| Rootstock | 101-14 | 31.20 | 96.05 [ab] | 1.09 [a] | 3.12 [ab] | 10.53 [ab] | 23.76 [a] | 3.15 [a] | 13.22 [a] |
| | 3309C | 34.31 | 104.16 [bc] | 1.11 [a] | 3.66 [bc] | 12.33 [bc] | 24.18 [a] | 3.09 [ab] | 13.06 [a] |
| | Own-rooted | 34.91 | 110.70 [c] | 1.25 [b] | 3.87 [c] | 13.06 [c] | 21.02 [b] | 3.05 [b] | 14.56 [b] |
| | Riparia Gloire | 32.35 | 99.59 [b] | 1.21 [b] | 3.36 [abc] | 11.33 [abc] | 23.27 [a] | 3.09 [ab] | 13.63 [a] |
| | SO4 | 34.76 | 87.64 [a] | 1.09 [a] | 3.10 [a] | 10.46 [a] | 23.51 [a] | 3.10 [ab] | 13.42 [a] |
| Year | 2015 | 26.08 [a] | 80.80 [a] | 1.03 [a] | 2.21 [a] | 7.45 [a] | 25.14 [a] | 3.44 [a] | 10.89 [a] |
| | 2016 | 36.58 [b] | 103.62 [b] | 1.27 [c] | 3.75 [c] | 12.62 [c] | 23.48 [b] | 3.06 [c] | 14.85 [b] |
| | 2017 | 24.53 [b] | 84.53 [a] | 1.09 [a] | 2.93 [b] | 9.87 [b] | 21.97 [c] | 3.10 [bc] | 11.45 [a] |
| | 2018 | 25.55 [a] | 106.78 [b] | 1.16 [b] | 2.73 [ab] | 9.20 [ab] | 23.48 [b] | 3.14 [b] | 14.09 [b] |
| | 2019 | 45.12 [c] | 123.18 [c] | 1.22 [bc] | 5.58 [d] | 18.79 [d] | 22.17 [c] | 2.82 [d] | 15.94 [c] |
| p-value | Rootstock | 0.1203 | <0.0001 | <0.0001 | 0.0003 | 0.0003 | <0.0001 | 0.0082 | <0.0001 |
| | Year | <0.0001 | <0.0001 | <0.0001 | <0.0001 | <0.0001 | <0.0001 | <0.0001 | <0.0001 |
| | Roostock × Year | 0.1246 | 0.1763 | 0.1875 | 0.3161 | 0.3161 | 0.6921 | 0.3680 | 0.9921 |

Values followed with a different letters in each column (section 'Rootstock' and 'Year', separately) were significantly different according to ANOVA.

Similarly to Frontenac, growing seasons affected yield components and fruit composition (Table 8). The years with the highest yield were 2019 and 2016, the lowest yield was collected during 2015, and the growing seasons 2017 and 2018 had intermediate yield. Yield may be related to the number of clusters, cluster weight, and berry weight which were higher during the 2019 and 2016 seasons compared to the other three growing seasons. Berry chemistry showed that the greatest maturity (expressed as a ratio of TSS/TA) was obtained during 2015 with the highest soluble solids level and the lowest titratable acidity value. During 2016 and 2018, we observed high sugar values but also high titratable acidity levels. In 2017, the level of soluble solids was low as was titratable acidity; during the 2019 growing season, we noted low soluble solid levels and very high values for titratable acidity.

### 3.4.3. Marquette

Of the studied varietals, Marquette was the most affected by grafting and growing season (Table 9). Statistical analyses demonstrated the greatest yield on own-rooted vines and those grafted on Riparia Gloire and 3309C. Yield per vine was directly related to the number of clusters that were higher for the same rootstocks. Cluster weight from vines grafted on 101-14 was lower than from vines grafted on Riparia Gloire. Berry weight varied from 1.19 to 1.37 g, with the heaviest berries on own-rooted vines and grafted on Riparia Gloire; the lightest berries were noted on rootstock 101-14. The highest productive years were 2016 and 2019, followed by 2017, 2015, and 2018 (Table 7). The number of clusters is related to yield obtained for each year: a higher number of clusters was observed during 2016, 2019, and 2017, and lower numbers during 2015 and 2018. Cluster weight varied from 80.80 g in 2015 to 123.18 g in 2019. Berries were heavier in 2016 and 2019 and lighter in 2017 and 2015.

The rootstocks also affected Marquette fruit composition at harvest. The soluble solid content was higher for vines grafted on rootstocks than for own-rooted vines. The titratable acidity was then linked to sugars, as we observed lower levels of titratable acidity on grafted vines compared to own-rooted. Berry maturity at harvest was also affected by growing season. The highest soluble solid concentration was noted in 2015; intermediate levels were observed during 2016 and 2018, and the lowest levels were found in 2019 and 2017. Titratable acidity was highest during the growing seasons 2019, 2016, and 2018, and lowest in 2017 and 2015.

**Table 9.** Effect of rootstock on yield quantity and fruit composition of grapevine 'Marquette'.

| | | | | | | | | | |
|---|---|---|---|---|---|---|---|---|---|
| | | **Marquette** | | | | | | | |
| | **Rootstock** | **Average Number of Clusters (pcs)** | **Cluster Weight (g)** | **Berry Weight (g)** | **Yield (kg·vine$^{-1}$)** | **Yield (t·ha$^{-1}$)** | **Soluble Solids (°Brix)** | **pH** | **Titratable Acidity (g·L$^{-1}$ tar. ac.)** |
| Rootstock | 101-14 | 29.98 [ab] | 60.68 [a] | 1.19 [a] | 1.86 [a] | 6.27 [a] | 24.61 [b] | 3.31 | 10.78 [ab] |
| | 3309C | 32.70 [a] | 71.02 [ab] | 1.27 [b] | 2.32 [c] | 7.83 [c] | 25.21 [ab] | 3.27 | 10.51 [a] |
| | Own-rooted | 31.35 [a] | 70.09 [ab] | 1.37 [c] | 2.20 [bc] | 7.40 [bc] | 24.95 [ab] | 3.24 | 11.20 [b] |
| | Riparia gloire | 31.45 [a] | 80.35 [b] | 1.35 [c] | 2.40 [c] | 8.07 [c] | 25.26 [a] | 3.23 | 11.21 [b] |
| | SO4 | 26.95 [b] | 71.74 [ab] | 1.32 [bc] | 1.99 [ab] | 6.70 [ab] | 23.45 [c] | 3.23 | 11.85 [c] |
| Year | 2015 | 26.37 [b] | 89.18 [a] | 1.14 [a] | 2.16 [b] | 7.26 [b] | 25.74 [a] | 3.72 [a] | 7.88 [a] |
| | 2016 | 36.63 [a] | 68.95 [b] | 1.44 [c] | 2.53 [a] | 8.53 [a] | 25.42 [a] | 3.16 [c] | 12.03 [c] |
| | 2017 | 33.97 [a] | 65.02 [bc] | 1.27 [b] | 2.22 [b] | 7.48 [b] | 24.39 [b] | 3.43 [b] | 9.01 [b] |
| | 2018 | 20.05 [c] | 58.25 [c] | 1.28 [b] | 1.26 [c] | 4.25 [c] | 23.72 [c] | 3.07 [d] | 12.58 [c] |
| | 2019 | 35.32 [a] | 75.16 [b] | 1.39 [c] | 2.66 [a] | 8.95 [a] | 24.45 [b] | 2.98 [e] | 13.45 [d] |
| *p*-value | Rootstock | <0.0001 | 0.0002 | <0.0001 | <0.0001 | <0.0001 | <0.0001 | 0.0972 | <0.0001 |
| | Year | <0.0001 | <0.0001 | <0.0001 | <0.0001 | <0.0001 | <0.0001 | <0.0001 | <0.0001 |
| | Roostock × Year | <0.0001 | 0.0003 | 0.6467 | <0.0001 | <0.0001 | 0.2392 | 0.1267 | 0.0936 |

Values followed with a different letters in each column (section 'Rootstock' and 'Year', separately) were significantly different according to ANOVA.

## 4. Discussion

Vines were planted in 2013, and results are more representative after a few years of production as grapevines became more established. The results obtained provide an interesting portrait of the impact of grafting on three cold-hardy hybrid varieties. The four rootstocks affected viticultural and physiological components, and some trends can be observed.

### 4.1. Bud Survival

Our results obtained for Frontenac, Frontenac blanc, and Marquette did not show a significant effect of rootstock on bud survival. The processes involved in bud survival and influenced by rootstock may be set by direct or indirect effects and could therefore also be related to factors such as vine growth, vine acclimation, and nutrient deficiency. The impact of rootstock on bud survival, however, is not consistent in the literature—some studies demonstrated an influence of rootstock on bud survival and others did not. Sabbatini and Howell [22] demonstrated that bud survival of grafted Vidal and Marechal Foch was influenced by the scion, and determined that no rootstock effect was observed. The same tendency was observed by Hoover et al. [6] with the cultivar St-Pepin grafted on five rootstocks. On the other hand, Striegler and Howell [12] observed that rootstocks could increase the bud survival of Seyval, with the most promising results coming from pairings with rootstock 3309 C. They noted the limited impacts of rootstock on bud survival and instead showed that, for some rootstocks, the vine size of grafted vines had a greater effect on bud survival. Hence, bud survival was influenced by indirect effects of grafting and was not directly related to the change in root system. These mixed findings are somewhat unsurprising given that commonly used rootstocks were developed to either mitigate soil biochemistry on vine development or enhance those same features. Rootstocks are not sought after nor principally known for their ability to impart freeze tolerance and improve cold hardiness of grapevines.

### 4.2. Grapevine Development

Vine physiology is causally linked to the characteristics of the scion, and the observations made throughout the trial period are in accordance with seasonal vine development for these three grape varieties. Howell [23] also noted that rootstocks did not impact

growth stages or seasonal vine development. Grafting had no effect on the phenology of interspecific hybrid vines under the conditions evaluated in this study.

The effect of rootstock on vine vigor showed that each variety was influenced differently by rootstock, but a tendency to lower vigor when grafted on Riparia Gloire was observed. Moreover, vigor was low for Frontenac blanc grafted on 101-14, and a weak vigor of Marquette was noted on rootstocks 3309C and 101-14. Rootstocks attribute a different influence on the vigor of grape varieties [9,10]. Following its establishment in 2013, rootstock 3309C was the one that generally produced a greater grapevine vigor, while vine grafted on 101-14 and Riparia Gloire rootstocks presented a reduced vigor. In subsequent growing seasons, where the vines were more established, it is difficult to draw a clear picture of the effects of rootstocks on the growth of the vines. According to the literature, rootstocks 3309C, 101-14, and SO4 are medium-vigor rootstocks, while Riparia Gloire is a low-vigor rootstock [9,19]. The study of Reynolds and Wardle [8] with interspecific hybrids also showed a variable effect of rootstock on vine vigor according to the combination of scion/rootstock. For example, De Chaunac and Marechal Foch grafted on Kober 5BB presented lower weight of cane pruning, and Seyval blanc showed lower pruning weight on own-root, SO4, and Kober 5BB. Hoover et al. [6] also observed an effect of rootstock on grapevine vigor. St-Pepin grapevine grafted on rootstock 3309 C and ES15-53 resulted in a heavier pruning weight than vine grafted on MN Rip 64 and MN 1065. On the other hand, Striegler and Howell [12] did not show a significant effect of rootstock on vine size and canopy development. Grapevines grafted on Seyval, Kober 5BB, and 3309 C had a similar vine size as the own-root vines.

### 4.3. Nutrient Deficiencies

The results demonstrate that magnesium deficiencies were more prevalent for the two Frontenac varieties compared to Marquette. Magnesium deficiency is frequently observed in vineyards, and several grape varieties even require higher magnesium uptake [17,24]. Overall, grafting improves the absorption of magnesium for several grape varieties, including the two Frontenac. Our results demonstrate that the most effective rootstocks for absorbing magnesium were 3309C and 101-14. These results are consistent with the descriptions of rootstocks found in the literature, where the rootstocks that are known to absorb magnesium more easily are 3309C and 101-14, while Riparia Gloire and SO4 rootstocks assimilate magnesium less easily [9,10].

### 4.4. Yield Parameters

Results related to yield components revealed significant differences between rootstocks and own-rooted vines. Frontenac was the least affected grape variety compared to Frontenac blanc and Marquette; only cluster weight and berry weight were impacted. For Frontenac blanc and Marquette, higher yields were observed on own-rooted vines and vines grafted on 3309C and Riparia Gloire. The number of clusters and cluster weight were generally higher for vines grafted on Riparia Gloire and 3309C, as well as on own-rooted vines. Other studies observed an effect of rootstocks on yield components for hybrid varieties [5,6,8,12,22,25]. Hoover et al. [6] demonstrated that there were few significant difference in yield and fruit composition among the scion/rootstock combinations for St-Pepin. However, Kaplan et al. [25], Harris [5], and Striegler and Howell [12] have shown a significant impact of rootstock on yield for Regent, Norton, and Seyval grape varieties, where higher yields were produced by vines grafted on Kober 5BB, 125AA, and 110R, respectively. Reynolds and Wardle [15] evaluated the effect of four rootstocks (compared to own-rooted vine) for nine hybrid varieties in British Columbia and the northwestern United States. Grapevine varieties studied were Chardonnay, Gewurztraminer, Ortega, Riesling, De Chaunac, Maréchal Foch, Okanagan Riesling, and Seyval blanc on rootstocks 3309C, 5BB, 5C, and SO4. For all scion/rootstock combinations, the results demonstrated weak-to-moderate effects of the rootstocks on yield. For example, grafting of De Chaunac, Okanagan Riesling, Gewurztraminer, and Riesling had no effect on yields or on the number

of clusters. Varieties most affected were Maréchal Foch and Chardonnay, where yields were higher on rootstock 5BB, while higher yields for Seyval blanc were observed for vines grafted on 3309C compared to SO4. The higher yield was mainly related to the number of clusters, which is higher per vine for these scion/rootstock combinations. Cluster weight is also sometimes affected by rootstock; however, results are often less correlated with yields than can be the number of clusters.

Yields observed for the three grape varieties were comparable to yields noted in other studies with these hybrid grape varieties under northeastern conditions [17,18,26–29]. The two Frontenac varieties are considered as productive varieties capable of averaging 10 to 12 t/ha, and growers may easily obtain higher yield with less consequent bud removal. Marquette is a little less productive and we generally reach between 6 to 8 t/ha. These three cold-hardy hybrids are found in many vineyards in eastern North America. In Quebec, Frontenac, Frontenac blanc, and Marquette are three grape varieties among the top 5 found in vineyards, representing 27.8% of the growing area with an estimated growing area of 534 acres [30]. Across the upper Midwestern United States and New England, 7580 acres out of a total of 55,500 acres are used for the production of cold-hardy hybrid grapes [31]. Marquette (24%) and Frontenac (17%) are two of the four most commonly planted cold-hardy hybrids grape in Minnesota, along with Frontenac gris and La Crescent [31,32].

### *4.5. Fruit Composition*

Chemical analysis demonstrated similar values for these hybrids in cold climates, soluble solids ranging from 20 to 26 °Brix, pH between 2.86 to 3.4, and titratable acidity from 10 to 20 g/L of tartaric acid for Frontenac and Frontenac blanc; between 22 to 30 °Brix, pH ranging from 2.84 to 3.5, and 8.2 to 13 g/L of tartaric acid for Marquette [18,33–36].

Overall, for the two Frontenac grape varieties, we observed a low soluble solid content, low pH, and a high titratable acidity for fruits on own-rooted vines compared to grafted vines on rootstocks 101-14 and 3309C. Grafting affected fruit composition for Marquette differently, where the lowest grape maturity was observed for fruits on vines grafted on SO4. Soluble solids and titratable acidity are significant indicators of grape ripening and fruit quality. Obtaining higher soluble solid content on grafted vines was also observed by Reynolds and Wardle [15] for Okanagan Riesling, Seyval blanc, and Chardonnay on rootstocks 5BB, SO4, and 3309C. Kaplan et al. [25] also observed an effect of rootstock on sugar content, where the lowest value was noted for rootstock 101-14 compared to others (SORI, 161-49C, 5BB, SO4, 125AA, and own-rooted). However, in several cases, grafting has shown no influence on the sugar content of juice at harvest, and this was found for several grape varieties (St. Pepin, De Chaunac, Maréchal Foch, Verdelet, Gewurztraminer, Riesling, Cabernet Franc, and Chardonnay) [6,7,15]. Similar results have been noted by other authors, where grafting has little or variable effects on the pH and titratable acidity of musts at harvest [6,7,15]. Regarding titratable acidity, rootstock affected all three grape varieties studies. Reynolds and Wardle [15] also noted a lower titratable acidity on grafted vines compared to own-rooted vines for the Okanagan Riesling and Verdelet grape varieties.

For the six years studied, we observed a seasonal effect on yield components and fruit composition. The 2015 growing season was the first harvest for the vines, and the low yield that season was coupled with a long and hot growing season (1429 GDD), resulting in grapes with high soluble solid content and low acidity. The most productive years were 2016 and 2019, with the highest yields and number of clusters; however, fruit composition was unbalanced, with a normal soluble solids content but high level of titratable acidity. The growing seasons of 2016 and 2019 were good growing seasons regarding GDD, but the high yield may have reduce the capacity of the vines to reduce fruit acidity [19]. Finally, the shortest growing season was observed during 2018, and we also noted more bud damage (not significant) for Marquette, resulting in a low yield with low sugar content and high acidity.

## 5. Conclusions

Grape rootstocks have been used in Europe since the end of the 19th century with *Vitis vinifera* to fight against phylloxera and nematode-infested soils or to adapt grapevines to specific soil conditions. However, although several studies have demonstrated the impact of rootstocks in improving vine performance and fruit composition, mainly for *V. vinifera*, the number of studies investigating their impact on hybrids grape varieties still remains very limited. This study has demonstrated that rootstocks may affect cold-hardy hybrids in different ways, and some of them showed higher potential than others for use in eastern North American conditions. The study of cold-hardy hybrids in eastern Canada is still relatively recent and more work needs to be done to improve knowledge about these grape varieties under specific growing conditions. Moreover, while rootstocks have since been shown to impart other adaptations to *V. vinifera*, the original use was never meant to impart cold-hardiness adaptations. Further research into the matter should test new and novel rootstocks that were developed specifically for and from North American hybrid cultivars.

**Author Contributions:** Conceptualization, C.P.; methodology, C.P.; validation, C.P.; formal analysis, C.P., A.C. and F.D.; investigation, C.P.; resources, C.P.; data curation, C.P. and A.C.; writing—original draft preparation, C.P.; writing—review and editing, C.P., A.C. and F.D.; supervision, C.P.; project administration, C.P.; funding acquisition, C.P. All authors have read and agreed to the published version of the manuscript.

**Funding:** Funding of this project has been provided in part through the AgriScience program-cluster on behalf of Agriculture and Agri-Food Canada.

**Institutional Review Board Statement:** Not applicable.

**Informed Consent Statement:** Not applicable.

**Data Availability Statement:** Not applicable.

**Acknowledgments:** The authors wish to thank Richard Bastien and Jérémie d'Hauteville for their expertise. We also thank Richard Kamal, Stefano Campagnaro, and Pascale Boulay for their technical support.

**Conflicts of Interest:** The authors declare no conflict of interest. The funders had no role in the design of the study; in the collection, analyses, or interpretation of data; in the writing of the manuscript; or in the decision to publish the results.

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
