# Peer review of "Rootstocks Impact Yield, Fruit Composition, Nutrient Deficiencies, and Winter Survival of Hybrid Cultivars in Eastern Canada"

_horticulturae, doi:10.3390/horticulturae7080237_

Round 1
Reviewer 1 Report
The manuscript titled „Rootstocks Impact Yield, Fruit Composition, Nutrient Deficiencies, and Winter Survival of Hybrid Cultivars in Eastern Canada” is about the first results of a long-term winegrape rootstock trial. The trial was planted in June of 2013, so the first some bearing years were described in this paper. The wine production is a bit challenging in the growing area due to its location.
Just a suggestion: please mark – may be under the title line – these results are the first results derived from this trial, so it needs more time, more bearing years to make the final conclusions about the observed rootstock/scion combinations.
Please give short descriptions about the rootstocks and scions observed in the trial to understand better the differences between the rootstock/scion combinations. Is it possible to give some detailed information about the soil?
It would be nice to mark the statistical groups with letters, which weren’t differed or differed from each other significantly to see the differences better.
Author Response
After all modifications made in the article, it is sent to English Editing Services. The final version edited will be added when it will be available.
Answer to reviewer comments:
Suggestion: this sentence was added at the beginning of the results section: The results presented are the first, and relevant conclusions will become in future with additional years of observations.
Descriptions of rootstocks and scions: information related to the four rootstocks and three scions were added to the Method section in Tables 1 and 2.
Soil description: in the Methods, this information was presented:
The vineyard soil type is a gravelly loam and plots are located on an 8% slope.
Statisticals groups: Letters are absent only in Table 5 and were added. All other Tables presented statistical differences with letters.

Reviewer 2 Report
In the presented manuscript the authors analyzed data from a 6-years project to identify the effects of grafting of different rootstock and vine cultivar combinations on cold resistance. For that a variety of important agronomic traits were measured and statistically analyzed. The results clearly show that different rootstock and vine cultivar combinations react differently to cold stress leading to increased or decrease resistance to cold. In general, the authors did a great job in statistically analyzing their data and comparing their results with the literature. However, I would like to point out a few things the authors should improve. I was a little bit surprised that the authors did not analyze the effects of precipitation, GDD, and days without frost with the agronomic data collected. The R-package ‘Vegan’ offers a lot of different functions which would help with that analysis. In addition, detailed comments are presented below.
Lines 28 – 45: These two paragraphs contain redundant information. The authors should combine them to reduce the redundancy.
Lines 47 – 51: These three sentences are redundant.
Lines 55 - 56: This sentence does not contain any new information and should be deleted, therefore.
Table 3: Please, add a short justification why LAI data were excluded for 2014 and 2015. The authors added level of significance to data in tables 4 to 7 but not in table 3. Was that on purpose? If so, why didn’t you add this information?
Table 4: What is the unit you measured the magnesium deficiency?
I recommend adding standard deviation to the data presented.
Author Response
After all modifications made in the article, it is sent to English Editing Services. The final version edited will be added when it will be available.
I was a little bit surprised that the authors did not analyze the effects of precipitation, GDD, and days without frost with the agronomic data collected. Other analyses were done regarding these parameters, but not relevant results were obtained; this is why we don’t include these analyses in this article.
Lines 28 – 45: These two paragraphs contain redundant information. The two paragraphs were combined to reduced redundant information.
Lines 47 – 51: These three sentences are redundant. Modifications were done to reduce information in only one sentence.
Lines 55 - 56: This sentence does not contain any new information and should be deleted, therefore. Sentence was removed
Table 3: Please, add a short justification why LAI data were excluded for 2014 and 2015. The authors added level of significance to data in tables 4 to 7 but not in table 3. Was that on purpose? If so, why didn’t you add this information? Letters were added to Table 3 (now Table 5). LAI data were not collected during the first two years of the study because the leaf area was too small to be managed with the equipment.
Table 4: What is the unit you measured the magnesium deficiency? This is a mean of Horsfall-Barratt grade scale.
I recommend adding standard deviation to the data presented. Adding the standard deviation data would make the tables more cumbersome, and it would not be easy to see the results correctly. That is why we did not present them.

Round 2
Reviewer 2 Report
Dear authors,
after reading the modified version of your manuscript, I think that it can be published in Horticulturae.